# Genomics of human and chicken *Salmonella* isolates in Senegal: Broilers as a source of antimicrobial resistance and potentially invasive nontyphoidal salmonellosis infections

**Yakhya Dieye**[1,2]*, **Dawn M. Hull**[3], **Abdoul Aziz Wane**[1], **Lyndy Harden**[3], **Cheikh Fall**[1], **Bissoume Sambe-Ba**[1], **Abdoulaye Seck**[1], **Paula J. Fedorka-Cray**[3], **Siddhartha Thakur**[3]

**1** Pole of Microbiology, Institut Pasteur, Dakar, Sénégal, **2** Département Génie Chimique et Biologie Appliquée, École Supérieure Polytechnique, Université Cheikh Anta Diop, Dakar, Sénégal, **3** Department of Population Health and Pathobiology, College of Veterinary Medicine, North Carolina State University, Raleigh, North Carolina, United States of America

* yakhya.dieye@pasteur.sn

**Data Availability Statement:** The paired-end reads used in this study were deposited at the National Center for Biotechnology Information (NCBI) under

## Abstract

*Salmonella enterica* is the most common foodborne pathogen worldwide. It causes two types of diseases, a self-limiting gastroenteritis and an invasive, more threatening, infection. *Salmonella* gastroenteritis is caused by several serotypes and is common worldwide. In contrast, invasive salmonellosis is rare in high-income countries (HIC) while frequent in low- and middle-income countries (LMIC), especially in sub-Saharan Africa (sSA). Invasive Non-typhoidal *Salmonella* (iNTS), corresponding to serotypes other than Typhi and Paratyphi, have emerged in sSA and pose a significant risk to public health. We conducted a whole-genome sequence (WGS) analysis of 72 strains of *Salmonella* isolated from diarrheic human patients and chicken meat sold in multipurpose markets in Dakar, Senegal. Antimicrobial susceptibility testing combined with WGS data analysis revealed frequent resistance to fluoroquinolones and the sulfamethoxazole-trimethoprim combination that are among the most used treatments for invasive *Salmonella*. In contrast, resistance to the historical first-line drugs chloramphenicol and ampicillin, and to cephalosporins was rare. Antimicrobial resistance (AMR) was lower in clinical isolates compared to chicken strains pointing to the concern posed by the excessive use of antimicrobials in farming. Phylogenetic analysis suggested possible transmission of the emerging multidrug resistant (MDR) Kentucky ST198 and serotype Schwarzengrund from chicken to human. These results stress the need for active surveillance of *Salmonella* and AMR in order to address invasive salmonellosis caused by nontyphoidal *Salmonella* strains and other important bacterial diseases in sSA.

the Bioproject accession numbers PRJNA293224. All other relevant data are within the paper and its Supporting information files.

**Funding:** The whole genome sequencing was completed by the FDA GenomeTrakr program funded grant 1U18FD00678801 (https://www.fda. gov/food/whole-genome-sequencing-wgs-program/genometrakr-network). YD was supported by The World Academy of Science (TWAS, http://www.twas.org) Grant No. 17-507 RG/BIO/AF/AC_G – FR3240300142. The funders had no role in study design, data collection and analysis, decision to publish, or preparation of the manuscript.

**Competing interests:** The authors have declared that no competing interests exist.

## Introduction

The modern human lifestyle characterized by frequent traveling and globalization of trade and exchange of goods, including food products, increases the rapid spread of infectious disease outbreaks. The current Covid-19 coronavirus crisis represents a dramatic example of the threat represented by infectious diseases in a globalized world. Addressing the problems caused by or related to infectious diseases in modern societies requires integrated, collaborative, and multi-disciplinary efforts. Surveillance systems based on a One Health approach play an essential role in monitoring pathogenic parasites, viruses, bacteria, and antimicrobial resistance (AMR) determinants relevant to human, animal, and environmental health.

*Salmonella enterica* is the most common foodborne pathogen worldwide. It comprises over 2600 serotypes [1] classified according to three major surface antigenic determinants [2]. *S. enterica* causes two types of disease: typhoid fever and nontyphoidal gastroenteritis. *Salmonella* gastroenteritis is restricted to the intestinal tract of the host [3]. Typhoid fever is more life-threatening and occurs when the bacteria invade the systemic compartment of the organism. Typhoid fever is caused by human specific serovars Typhi and Paratyphi A. As for many infectious diseases, the incidence, prevalence, and outcome of *Salmonella* infections drastically differ between rich and poor regions of the globe. While *Salmonella* gastroenteritis is still a concern in both high-income (HIC) and low- and middle-income countries (LMIC), typhoid fever cases are very rare in the former. Additionally, it is known that nontyphoidal *Salmonella* (NTS) serovars can potentially cause systemic infections termed invasive nontyphoidal salmonellosis (iNTS) [4]. Typhoid fever and iNTS are estimated to cause over 25 million cases and nearly 900,000 deaths annually [5]. In sSA, several multi-country surveillance programs [6] as well as sporadic studies [7–9] have generated a wealth of epidemiological data on the prevalence and incidence of typhoidal and iNTS. Available data indicate that iNTS is a significant cause of invasive disease in sSA, with Typhimurium and Enteritidis being the most frequently isolated serovars [5, 10, 11]. This is of noticeable importance given that these serotypes can infect multiple hosts, and are transmissible to humans through various sources, including food, animals, and the environment. Food animals (poultry, cattle and pigs) and their products (including eggs, milk and pork) constitute important sources of *Salmonella* contamination [12–15]. These animals are healthy asymptomatic carriers of many *Salmonella* serotypes, which can cause clinical disease in humans. These carriers serve as reservoirs for dissemination of not only the bacteria, but also of the antimicrobial resistance (AMR) determinants they harbor.

AMR is a global public health threat. Multidrug resistant (MDR) clones of important human pathogens are emerging at an alarming rate, and experts agree that if proper actions are not taken, the number of deaths due to resistant bacteria will increase in the future. Dissemination of AMR is due to several causes and influenced by adaptive pressures instituted from antimicrobial usage. The emergence of antimicrobial resistant *Salmonella* strains was first reported in the 1950s, soon after the introduction of chloramphenicol for the treatment of salmonellosis [16]. Since then, the prevalence of resistant clones in humans, domestic animals, and other wildlife species have expanded globally [17]. MDR clones of *Salmonella*, defined as co-resistance to the traditional first-line drugs chloramphenicol, ampicillin, and sulfamethoxazole-trimethoprim [18, 19], started emerging in the 1980s, necessitating the use of the second-line drugs fluoroquinolones and third-generation cephalosporins [20]. The selective pressure exerted by inappropriate use of antimicrobials, particularly as growth-promoting biologics for farm animals in countries where there is no regulation banning this practice, has led to the emergence of extensively drug resistant (XDR) clones defined as non-susceptibility to at least one agent in all but two or fewer antimicrobial categories [21]. For example, the XDR

Kentucky serovar ST198 emerged in Egypt ~1989 following the acquisition of a genomic island harboring several AMR genes and spread widely throughout the world, acquiring additional resistance-conferring mutations [22]. Kentucky ST198 was associated with clinical infections in humans in different regions of the globe [22]. Currently, there is an ongoing epidemic of XDR serovar Typhi that emerged in Pakistan, which has been introduced to the USA by individuals returning from this affected region [23]. These emerging MDR and XDR clones of *Salmonella* represents a serious public health threat, especially in sSA, where there is a lack of laboratories suitably equipped to perform susceptibility testing. Consequently, when available, antimicrobial therapy is often empirical, which could promote the selection of resistant clones. Surveillance systems that monitor AMR are scarce in sSA. Available data indicate that resistance to first-line, cheap antimicrobials emerged in the 1980s [24]. Recent reports described emergence of MDR clones, which express resistance to the most efficient antimicrobials against *Salmonella*, such as third-generation cephalosporins and fluoroquinolones [20, 24]. In Senegal, strains resistant to antimicrobials used for treatment of salmonellosis, including MDR clones, have been isolated from human, food, and animal samples [14, 15, 25, 26]. In this study, we performed whole genome sequencing of 72 *Salmonella* isolates recovered from humans (n = 19) and chicken meat (n = 53) during a surveillance project in Dakar, Senegal [14]. The aim of our study was to determine the resistance profile of nontyphoidal *Salmonella* strains from clinical salmonellosis cases in humans, and the potential transmission of these strains from retail chicken meat. We report on the analysis of serotype distribution, phylogenic relationship, phenotypic and genotypic AMR profiles, and virulence and plasmid composition of isolates from the two sources analyzed.

## Materials and methods

### Sampling, culture, and identification of *Salmonella* isolates

*Salmonella* isolates from humans were recovered from diarrheic stools of patients visiting the clinical microbiology laboratory of the Institut Pasteur de Dakar, Senegal between July 2012 and June 2013. Isolates from chicken were recovered, during the same period, from carcasses sold at multipurpose markets in Dakar, Senegal. Carcasses were placed in sterile bags with ice packs, immediately sent to the laboratory for microbiological analysis, and processed as previously described [14]. Briefly, 25 mg of mixed parts of the chicken carcasses (part of the neck, thighs, anus, and skin) were removed, ground, mixed with 250 ml of buffered peptone water, and incubated at 37˚C overnight. Then, 0.1 ml of the sample was used to inoculate 10 ml of Rappaport-Vassiliadis enrichment medium for *Salmonella* and incubated at 42˚C for 24 hours. Dilutions of the cultures were spread onto XLD and Hektoen plates. *Salmonella* colonies were recovered and confirmed by biochemical testing (Api 20E, ref 20100; Biomerieux, France). Human and chicken isolates were stored in glycerol stocks until used in further biological analysis.

### Antimicrobial susceptibility testing

Susceptibility to antimicrobial drugs was tested using a standard disc diffusion method on Mueller–Hinton agar plates (Oxoid) and using diameters defined by the Clinical and Laboratory Standards Institute (CLSI). The following 22 antimicrobials (or combination of antimicrobials) were tested: chloramphenicol (Cm), sulfamethoxazole + trimethoprim (ST), ampicillin (Ap), ticarcillin (Tc), amoxicillin + clavulanic acid (AC), aztreonam (At), cefalothin (Cf), cefoxitin (Cx), cefotaxime (Ct), ceftazidime (Cz), cefepime (Fp), imipenem (Ip), gentamicin (Gm), kanamycin (Kn), tobramycin (Tm), nalidixic acid (Na), ciprofloxacin (Cp), norfloxacin (No), ofloxacin (Ox), tetracycline (Te), colistin (Co), and erythromycin (Er).

## Extraction of bacterial genomic DNA, whole genome sequencing, and assembly

Bacterial genomic DNA was extracted using the MasterPure™ Gram Positive DNA Purification Kit according to the recommendation of the manufacturer. This kit was used because, in our hands, it consistently yielded *Salmonella* genomic DNA of better quality for sequencing on an Illumina platform. The quality and concentration of the extracted DNA were determined using a NanoDropTM 2000/2000c Spectrophotometer (Thermo Fisher Scientific, Waltham, Massachusetts) and a Qubit 3.0 Fluorometer (Thermo Fisher Scientific). DNA libraries were prepared using the Nextera XT DNA Library Preparation Kit (Illumina, San Diego, CA) following the manufacturer's instructions. The resulting DNA libraries were purified using AMPure XP beads (Beckman Coulter, Sharon Hill, PA) and re-quantified using the Qubit 3.0 Fluorometer (Thermo Fisher Scientific). Sequencing was performed on the MiSeq System using v2 sequencing reagent kits (Illumina).

Whole genome sequence forward and reverse reads were assembled *de novo* with Shovill v1.1.0 using SPAdes v3.14.1 [27, 28]. *De novo* assembly of forward and reverse reads was repeated for plasmid detection using plasmidSPAdes genome assembler v3.14.1 [29]. Plasmid contigs (in fasta file format) were further analyzed for antimicrobial resistance genes, virulence factors, and plasmid replicons using Abricate 1.0.1 [30]. Abricate databases include NCBI AMRFinderPlus (doi:10.1128/AAC.00483-19), Comprehensive Antibiotic Resistance Database (CARD) (doi:10.1093/nar/gkw1004), Resfinder (doi:10.1093/jac/dks261), Virulence Factor Database (VFDB) (doi:10.1093/nar/gkv1239), PlasmidFinder (doi:10.1128/AAC.02412-14), and MEGARES 2.00 (doi:10.1093/nar/gkz1010). Genes detected with a minimum 90% coverage and identity were reported.

## Serotype prediction, detection of antimicrobial resistance genes, and phylogenetic analysis

*Salmonella* serotypes were predicted by submitting the contigs from genome assembly to the SeqSero 2 (http://denglab.info/SeqSero2) platform [31]. Sequence types were determined using the MLST 2.0 software [32]. Antimicrobial resistance genes and alleles with a minimum of 90% coverage and identity were detected from assembled contigs using ResFinder 4.1 software [33, 34] at the Center for Genomic Epidemiology (www.genomicepidemiology.org). For phylogenetic analysis, compiled genome assembly SNP alignments were constructed using CSI Phylogeny 1.4 from the Center for Genomic Epidemiology [35] with all parameters set at default. Generated phylogenetic tree was visualized using R Studio software (http://www.rstudio.com/).

## Results

### Demographic data

We performed WGS of 72 *Salmonella* isolates that represented a subset of strains collected during a surveillance project between July 2012 and June 2013 in Senegal. The isolates included 19 and 53 strains originating from human and retail chicken meat, respectively (Table 1, see Materials and methods).

### Serotype distribution

Analysis of the WGS data revealed 24 different serotypes, the most represented being Brancaster (14 isolates), Kentucky (13 isolates), Hadar (11 isolates), Chester (four isolates), Schwarzengrund (four isolates), and Senftenberg (four isolates) (Table 1). Serotype distribution within

**Table 1. Serotypes and origin of *Salmonella* isolates.**

| Serotypes | Human | Chicken | Total |
|---|---|---|---|
| Brancaster | 0 | 14 | 14 |
| Kentucky | 1 | 12 | 13 |
| Hadar | 0 | 11 | 11 |
| Chester | 0 | 4 | 4 |
| Schwarzengrund | 1 | 3 | 4 |
| Senftenberg | 0 | 4 | 4 |
| Banana | 3 | 0 | 3 |
| Gaminara | 2 | 0 | 2 |
| Johannesburg | 0 | 2 | 2 |
| Isangi | 1 | 0 | 1 |
| Give | 1 | 0 | 1 |
| Poona | 1 | 0 | 1 |
| Corvallis | 1 | 0 | 1 |
| Somone | 1 | 0 | 1 |
| Muenster | 1 | 0 | 1 |
| Baildon | 1 | 0 | 1 |
| Oranienburg | 1 | 0 | 1 |
| 3,10:e,h:- | 1 | 0 | 1 |
| Virchow | 1 | 0 | 1 |
| Rissen | 1 | 0 | 1 |
| Okerara | 1 | 0 | 1 |
| Typhimurium | 0 | 1 | 1 |
| Brandenburg | 0 | 1 | 1 |
| Vejle | 0 | 1 | 1 |
| **Total** | **19** | **53** | **72** |

groups showed that isolates from humans were diverse, with 16 serotypes for 19 strains (Table 1). Only serotype Banana (three isolates) and Gaminara (two isolates) were found in more than one human sample. Interestingly, the Banana isolates were recovered in October 2012 (two isolates) and July 2013 (one isolate), and the two Gaminara isolates in September and November 2013, suggesting a possibility of diarrheic outbreaks, likely of low intensity, caused by these serotypes. Additionally, all but two (Kentucky and Schwarzengrund) of the serotypes from humans were not found in chicken. A phylogenetic analysis showed that the serotypes from stools were genetically diverse, confirming the ability of various *Salmonella* serovars to cause diarrheal disease in humans (Fig 1). Contrary to strains from humans, there was less serotype diversity in isolates from chicken meat, with 10 different serovars for 53 strains (Table 1). Serotypes Brancaster (14 isolates), Kentucky (12 isolates), and Hadar (11 isolates) predominated among these bacteria.

## Phylogenetic analysis of *Salmonella* isolates from human and chicken

A concatenated SNP-based phylogeny was performed to evaluate the relationship among the 72 strains. Not surprisingly, the isolates belonging to the same serotypes mostly clustered together (Fig 1). The only exception was serotype Kentucky, whose 13 members were split in two distinct clusters of nine and four isolates, respectively (Fig 1). Isolates of each of the two clusters shared the same sequence types (ST) that were ST314 and ST198 for the first and

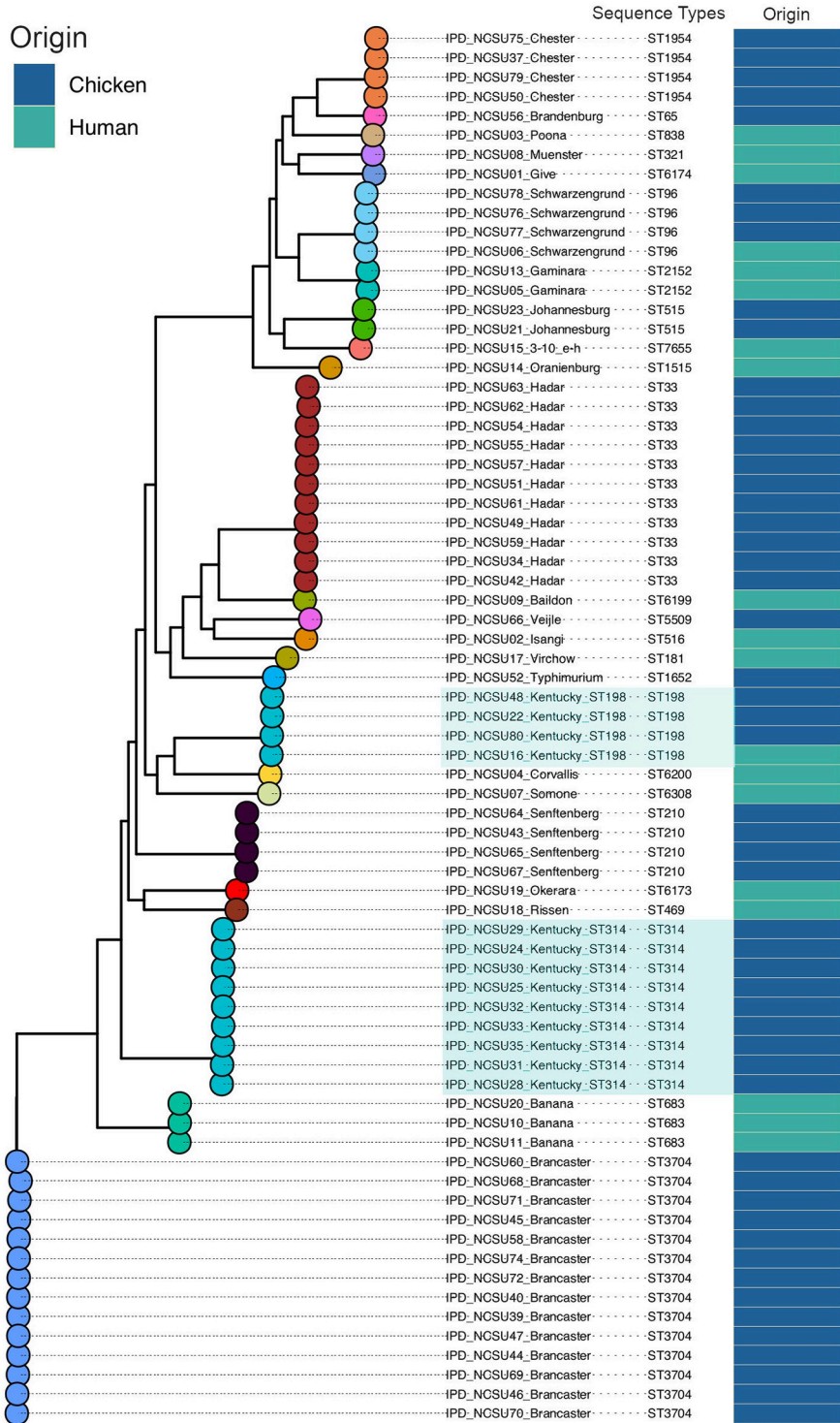

**Fig 1. Phylogenetic analysis of *Salmonella* strains from human and chicken.** SNPs from genome assemblies were compiled and used to build alignment and generate a phylogenetic tree CSI Phylogeny 1.4 with all parameters set at default. The generated tree was visualized using R Studio. Isolates belonging to Kentucky sequence types ST198 and ST314 that formed two distinct clades are shaded in blue.

second clades, respectively. Serotype Kentucky of the ST198 represents a lineage that recently emerged following the acquisition of a genomic island harboring multidrug resistant determinant [22] and disseminate throughout the world (see below). Interestingly, there were two phylogenetic clades composed of the four Kentucky ST198 in one hand and the four isolates of serotype Schwarzengrund in the other, which contained each, human and chicken isolates (Fig 1), suggesting a possible transmission to humans of these bacteria, likely through consumption of chicken meat.

## Antimicrobial resistance profile of *Salmonella* isolates

All isolates were tested for susceptibility to 22 antimicrobials belonging to 11 classes (Table 2). Resistance to the historical first line *Salmonella* treatment was rare for chloramphenicol (1.4%, [1/72]) and ampicillin (4.2%, [3/72]), and frequent for the trimethoprim-sulfamethoxazole anti-folate combination (38.9%, [28/72]), which was mostly isolated from chicken (27/28) with only one human isolate displaying this phenotype (Table 2). Regarding the second-line drugs for *Salmonella* treatment, resistance to cephems was rare, with only one human isolate resistant to first (cefalothin), second (cefoxitin), and third-generation (cefotaxime and ceftazidime) cephalosporins, while all the tested isolates were sensitive to cefepime that belongs to the fourth-generation of this drug class and to imipenem, a carbapenem. In contrast, resistance to fluoroquinolones was frequent (19.4%, [14/72] for ciprofloxacin; 18.1%, [13/72] for ofloxacin) except for norfloxacin (4.2%, [3/72]) (Table 2). Besides these antimicrobials, the drugs for which the isolates were most frequently resistant were tetracycline (48.6%; [35/72]) and

**Table 2. Resistance of human and chicken isolates of *Salmonella* to different classes of antimicrobials.**

| Classes | Antimicrobials | Origin and number of resistant isolates | | Total |
|---|---|---|---|---|
| | | Human | Chicken | |
| Phenicol | Chloramphenicol | 1 | 0 | 1 |
| Anti-folate | Sulfamethoxazol +Trimethoprim | 1 | 27 | 28 |
| Penicillin | Ampicillin | 2 | 1 | 3 |
| | Ticarcillin | 2 | 1 | 3 |
| | Amoxicillin + Clavulanic Acid | 1 | 0 | 1 |
| Monobactam | Aztreonam | 1 | 0 | 1 |
| Cephem | Cefalothin | 1 | 0 | 1 |
| | Cefoxitin | 1 | 0 | 1 |
| | Cefotaxime | 1 | 0 | 1 |
| | Ceftazidime | 1 | 0 | 1 |
| | Cefepime | 0 | 0 | 0 |
| Carbapenem | Imipenem | 0 | 0 | 0 |
| Aminoglycoside | Gentamicin | 2 | 3 | 5 |
| | Kanamycin | 0 | 2 | 2 |
| | Tobramycin | 1 | 0 | 1 |
| Quinolone | Nalidixic Acid | 1 | 14 | 15 |
| | Ciprofloxacin | 1 | 14 | 15 |
| | Norfloxacin | 1 | 2 | 3 |
| | Ofloxacin | 1 | 12 | 13 |
| Cycline | Tetracycline | 1 | 35 | 36 |
| Polymixin | Colistin | 0 | 0 | 0 |
| Macrolide | Erythromycin | 2 | 22 | 24 |

erythromycin (30.6%, [22/72]). No resistance was observed for imipenem, and one isolate displayed resistance to aztreonam (Table 2).

Multidrug resistance, defined in *Salmonella* as co-resistance to the historical first-line drugs including chloramphenicol, ampicillin, and trimethoprim-sulfamethoxazole combination [36] was found in only one strain, a human isolate of serotype Poona that was resistant to 13/22 antimicrobials tested (Table 1). However, 27.8% (20/72) of the isolates displayed resistance to at least three antimicrobial classes, a common definition of MDR phenotype [37].

Interestingly, AMR was more frequent in isolates from chicken. Indeed, 89.5% (17/19) of the human isolates were sensitive to the 22 antimicrobials tested, while only 17% (9/53) of the chicken strains displayed this feature (Table 1). This observation points to the selection pressure exerted by the use of antimicrobials as a growth promoter in chicken farming.

## Whole-genome sequencing (WGS) analysis

Twenty-seven known antimicrobial resistance genes (ARG) and four point mutations previously described as conferring resistance to quinolone, including two on *parC* and two on *gyrA* genes, were detected from the WGS data (Table 3). The ARGs encoding resistance to different classes of AMR factors, including aminoglycoside inactivating enzymes, anti-folates, β-lactamases, quinolone resistance proteins, rifamycin inactivator, fosfomycin inhibitor, and efflux pumps, were detected.

Genes encoding aminoglycoside modifying enzymes were the most frequently found ARG with 10 genes detected in 34 isolates. However, four ARG belonging to the phosphotransferase subfamily (*aph(3')-Ia*, *aph(3')-VIa*, *aph(3")-Ib*, *aph(6)-Id*) were not associated with any resistance. Similarly, one aminoglycoside acetyltransferase ARG (*aac(6')-Iaa*) detected in all the isolates without conferring resistance, is a cryptic gene present in most *Salmonella* isolates [38]. The remaining aminoglycoside modification ARG included the nucleotidyltransferases *aadA1*, *aadA2*, and *ant(2')-Ia* that were present together in a MDR human isolate of serotype Poona that displayed resistance to gentamycin and tobramycin. Additionally, *aac(3)-Id* (acetyltransferase) and *aadA7* (nucleotidyltransferase) were co-detected in three gentamicin-resistant isolates of serotype Kentucky including two and one of sequence types ST198 and ST314 respectively (Table 3).

Six ARG targeting folate pathways were detected, including *sul1* and *sul2* that confer resistance to sulfonamides, and four genes of the *dfrA* family that confer resistance to trimethoprim. Of the 28/72 isolates displaying resistance to the sulfamethoxazole-trimethoprim combination, the presence of at least one member of each of these two anti-folate subfamilies was detected. However, three isolates belonging to serotype Kentucky ST314 (n = 2) and Hadar (n = 1) possessed these genes without displaying resistance to sulfamethoxazole-trimethoprim. Additionally, one isolate of Kentucky ST198 was resistant to sulfamethoxazole-trimethoprim while possessing a *sul1* only without a *dfrA* gene (Table 3).

Three ARGs of the quinolone resistance family and four point mutations in *parC* and *gyrA* were detected in the isolates. One *parC* mutation (ParC T57S) was found in 69/72 isolates but was not associated with phenotypic quinolone resistance. All the 16/72 strains resistant to at least one quinolone possessed at least one *qnrB* ARG and/or a point mutation in *parC* and/or *gyrA*. Regarding other antimicrobials relevant to *Salmonella* treatment, one human isolate of serotype Poona resistant to chloramphenicol possessed a *floR* ARG as expected (Table 3). Similarly, four isolates displaying resistance to a β-lactamase harbored a corresponding gene. The extended spectrum β-lactamase (ESBL) *blaOXA-10* and *blaTEM-1* ARGs conferred resistance to the penicillin family (ampicillin, ticarcillin, amoxicillin+ clavulanic acid), while *blaDHA-1* and *blaCMY-2* provided resistance to first, second, and third classes cephalosporins (Table 3).

**Table 3. Antimicrobial resistance profile and resistance gene content of 72 human and chicken isolates of *Salmonella*.**

| Serotypes | AMR Phenotypic Profile | ARG[&] | Plasmidic ARG |
|---|---|---|---|
| Brancaster | (Kn) Na Cp Ox Te Er | *sul2 tetB aph(3')-Ib aph(3")-Ib aph(6)-Id qnrB19* | *sul2 tetB qnrB19 tetD* |
| Brancaster | (Kn) Na Cp Ox Te Er | *sul2 tetB aph(3')-Ib aph(3")-Ib aph(6)-Id qnrB6* | *qnrB19* |
| Brancaster (X 4)[#] | ST Te | *sul2 dfrA1 tetB aph(3')-Ib aph(3")-Ib aph(6)-Id* | |
| Brancaster (X 4)[#] | ST Te | *sul2 dfrA1 tetB aph(3')-Ib aph(3")-Ib aph(6)-Id* | *sul2 dfrA1 aph(3")-Ib aph(6)-Id* |
| Brancaster | Kn (Tm) Te Er | *sul2 tetB aph(3')-Ib aph(3")-Ib aph(6)-Id* | |
| Brancaster | ST (Kn) Na Cp Te Er | *sul2 dfrA1 dfrA15 tetA tetB aph(3')-Ib aph(3")-Ib aph(6)-Id qnrB19* | *sul2 dfrA1 tetA aph(3")-Ib aph(6)-Id qnrB19* |
| Brancaster | Te | *sul2 tetB aph(3')-Ib aph(3")-Ib aph(6)-Id* | |
| Brancaster | (Kn) Te Er | *sul2 tetB aph(3')-Ib aph(3")-Ib aph(6)-Id* | |
| Kentucky ST198 | Ap Tc Gm (Tm) Na Cp No Ox Te Er | *aac(3)-Id aadA7 tetA sul1 blaTEM-1b parC-S80I gyrA-S83F, gyrA-D87N* | |
| Kentucky ST198 | ST Na Cp No Ox Te Er | *sul2 dfrA14 tetA aph(6)-Id parC-S80I* | *sul2 dfrA14 tetA aph(6)-Id* |
| Kentucky ST198 | ST Ap Tc AC Gm Na Cp No Ox Te Er | *aac(3)-Id aadA7 blaTEM-1B sul1 tetA parC-S80I gyrA-S83F, gyrA-D87N* | |
| Kentucky ST198 | ST Na Cp Ox Te Er | *sul2 dfrA14 tetA aph(6)-Id parC-S80I gyrA-S83F, gyrA-D87N* | *sul2 dfrA14 tetA aph(6)-Id* |
| Kentucky ST314 (X 2)[#] | Er | **sul1 dfrA15**[*] | |
| Kentucky ST314 | ST Gm Te Er | *aac(3)-Id aadA7 sul1 dfrA15 tetA aph(3")-Ib aph(6)-Id* | |
| Kentucky ST314 (X 6)[#] | ST | *sul1 dfrA15* | |
| Hadar (X 2)[#] | ST Na Cp (Ox) Te Er | *sul2 dfrA1 tetA aph(3")-Ib aph(6)-Id qnrB19* | *sul2 dfrA1 aph(3")-Ib aph(6)-Id qnrB19* |
| Hadar | Te Er | **sul2 dfrA1**[*] *tetA aph(3")-Ib aph(6)-Id* | |
| Hadar (X 6)[#] | Te | *tetA aph(3")-Ib aph(6)-Id* | |
| Hadar | ST Gm (Tm) Na Cp Ox Te Er | *sul2 dfrA1 tetA aph(3")-Ib aph(6)-Id qnrB19* | |
| Hadar | Te | *tetA aph(3")-Ib* | |
| Chester | ST (Kn) Na Cp Ox Te Er | *sul2 dfrA14 tetA aph(6)-Id qnrB19* | |
| Chester | sensitive | | |
| Chester | ST Na Cp Ox Te Er | *sul2 dfrA14 tetA aph(6)-Id qnrB19* | *sul2 dfrA1 tetA aph(6)-Id* |
| Chester | ST Na Cp Ox Te Er | *sul2 dfrA14 tetA aph(6)-Id qnrB19* | |
| Schwarzengrund | sensitive | | |
| Schwarzengrund (X 2)[#] | sensitive | | |
| Schwarzengrund | Er | | |
| Senftenberg (X 4)[#] | sensitive | | |
| Banana (X 3)[#] | sensitive | *fosA* | |
| Gaminara (X 2)[#] | sensitive | | |
| Johannesburg | ST Na Cp Ox Te Er | *sul1 dfrA7 tetA gyrA-S83F* | |
| Johannesburg | ST Kn Na Cp Ox Te Er | *sul1 dfrA7 tetA gyrA-S83F* | |
| Isangi | sensitive | | |
| Give | sensitive | | |
| Poona | Cm ST Ap Tc AC At Cf Cx Cz Ct Gm Tm Er | *aadA1 aadA2b ant(2")-Ia aph(3')-Ia aph(3')-VIa arr-2 sul1 sul2 floR blaDHA-1 blaOXA-10 dfrA14* | *aph(3')-Ia aph(3')-VIa sul1 sul2 floR blaDHA-1* |
| Corvallis | sensitive | | |
| Somone | sensitive | *fosA* | |
| Muenster | sensitive | | |
| Baildon | sensitive | | |

*(Continued)*

**Table 3.** (Continued)

| Serotypes | AMR Phenotypic Profile | ARG [&] | Plasmidic ARG |
|---|---|---|---|
| Oranienburg | sensitive | *fosA* | |
| 3,10:e,h:- | sensitive | *fosA* | |
| Virchow | sensitive | | |
| Rissen | sensitive | | |
| Okerara | sensitive | | |
| Typhimurium | Ox Te Er | *fosA tetA qnrB7* | *tetA qnrB7* |
| Brandenburg | sensitive | | |
| Vejle | sensitive | | |

ARG, antimicrobial resistance gene. Antimicrobials: AC, amoxicillin + clavulanic acid; Ap, ampicillin; At, aztreonam; Cf, cefalothin; Cx, cefoxitin; Ct, cefotaxime; Cz, ceftazidime; Cm, chloramphenicol; Cp, ciprofloxacin; Er, erythromycin; Fp, cefepime; Co, colistin; Gm, gentamicin; Ip, imipenem; Kn, kanamycin; Na, nalidixic acid; No, norfloxacin; Ox, ofloxacin; ST, sulfamethoxazole + trimethoprim; Tc, ticarcillin; Te, tetracycline; Tm, tobramycin.

[&], gene *aac(6')-Iaa*, not shown in Table 3, was present in all isolates without conferring AMR

[*], *sul* and *dfrA* genes present in isolates susceptible to sulfamethoxazole + trimethoprim.

[#], Number of isolates that have the same resistance phenotype and ARG profile.

Isolates from humans are shaded.

As expected, of the 36/72 tetracycline resistant isolates, 23 and 14 possessed a *tetA* or a *tetB* efflux pump respectively, while one harbored both. Noticeably, while we did not test phenotypic resistance to fosfomycin, seven isolates (six human and one chicken) harbored a *fosA7* gene conferring resistance to this antimicrobial (Table 3).

## Plasmid presence and content in *Salmonella* isolates

PlasmidSPAdes analysis detected putative plasmids in 55.6% (40/72) of the isolates, including 42.1% (8/19) and 60.4% (32/53) of the human and chicken strains, respectively (S1 Table). Plasmid presence tended to match with serotypes and was more frequent among the serotypes Brancaster (64.3%, [9/14]), Hadar (81.8%, [9/11]), Senftenberg (100%, [4/4]), and Schwarzengrund (75%, [3/4]) (S1 Table). Noticeably, all (4/4) the Kentucky isolates of sequence type ST198 harbored plasmids while those of ST314 (9/9) were devoid of a plasmid. The ColRNAI type was the most frequently found replicon (19 occurrences), followed by Col156 (10 occurrences) and ColpVC (nine occurrences), while different Inc types were found at low frequency (1 to 4) (S1 Table).

Plasmids containing ARGs were found in 15 genomes, including a single clinical strain of serotype Poona and 14 chicken isolates of serotypes Brancaster (n = 7), Kentucky ST198 (n = 2), Hadar (n = 2), Chester (n = 1), Schwarzengrund (n = 1), and Typhimurium (n = 1). Gene *sul2* was the most frequent plasmidic ARG, being present in 13/15 isolates with this feature (Table 4). Interestingly, *sul2* occurred with a *drfA1* or *dfrA14* genes in 12/15 cases suggesting a co-selection of determinant conferring resistance to the trimethoprim-sulfonamide antifolate combination. Of note, the *blaDHA1* and *blaCMY-2* ARGs conferring resistance to first, second and third class cephalosporins were located on putative IncA/C and IncI1plamids, respectively (Table 4).

Besides ARGs, other relevant plasmid content interestingly included virulence genes of pathogenic *E. coli* (Table 4). One clinical isolate of serotype Gaminara harbored an IncI1 type plasmid encoding the Pic serine protease autotransporter found in enteroaggregative *E. coli* (EAEC) and in *Shigella* [39], and the EAST1 toxin found in various pathogenic *E. coli* and in some *Salmonella* [40] (Table 4). Another isolate of the same serotype harbored an IncFII type

**Table 4. Antimicrobial resistance and virulence genes found on plasmids of human and clinical isolates of *Salmonella*.**

| Serotypes | Plasmid replicons | ARG | Virulence genes |
|---|---|---|---|
| Brancaster | Col440I | *sul2 tetB qnrB19 tetD* | *papI, papB* |
| Brancaster | Col440I | *qnrB19* | |
| Brancaster | ColRNAI | *sul2 dfrA1 aph(3")-Ib aph(6)-Id* | |
| Brancaster | ColRNAI / Col440I<br>Col156<br>ColpVC | *sul2 dfrA1 tetA aph(3")-Ib aph(6)-Id qnrB19* | |
| Brancaster | ColRNAI | *sul2 dfrA1 aph(3")-Ib aph(6)-Id* | |
| Brancaster | ColRNAI | *sul2 dfrA1 aph(3")-Ib aph(6)-Id* | |
| Brancaster | ColRNAI | *sul2 dfrA1 aph(3")-Ib aph(6)-Id* | |
| Kentucky ST198 | ColRNAI / Col440II | *sul2 dfrA14 tetA aph(6)-Id* | |
| Kentucky ST198 | ColRNAI / Col440II | *sul2 dfrA14 tetA aph(6)-Id* | |
| Hadar | ColRNAI / Col440II Col156 | *sul2 dfrA1 aph(3")-Ib aph(6)-Id qnrB19* | |
| Hadar | ColRNAI / Col440I | *sul2 dfrA1 aph(3")-Ib aph(6)-Id qnrB19* | |
| Chester | ColRNAI / Col440I Col156 | *sul2 dfrA1 tetA aph(6)-Id* | |
| Schwarzengrund | IncI2<br>IncFIB<br>IncP1 | *sul2 tetA aph(3')-Ia aph(3")-Ib aph(6)-Id* | *papI, papB, papH, papDJKEFG* |
| Typhimurium | IncX3 / IncY<br>ColRNAI / Col440 | *tetA qnrB7* | |
| Poona | IncA/C | *aph(3')-Ia aph(3')-VIa sul1 sul2 floR blaDHA-1* | |
| Gaminara | IncI1 | ND | *pic, astA* |
| Gaminara | IncFII | ND | *faeEDC* |
| Johannesburg | ColRNAI / Col440II ColpVC | ND | *papDJKEFGC* |

ARG, antimicrobial resistance gene; *pap*, pyelonephritis-associated pili; (*pic*) serine protease precursor [Pic (VF0232)] [*Escherichia coli* CFT073]; (*astA*) heat-stable enterotoxin 1 [EAST1 (VF0216)] [*Escherichia coli* O44:H18 042]; *fae*, gene encoding a component of the K88 fimbriae.

ND, no ARG detected.

Isolates from humans are shaded.

plasmid with the *faeC*, *faeD*, and *faeE* genes that are part of an operon encoding K88 fimbriae associated with virulence in swine of enterotoxigenic *E. coli* (ETEC) [41] (Table 4). Additionally, plasmids harboring genes of the pyelonephritis-associated pili (PAP) of uropathogenic *E. coli* (UPEC) [42] were detected in three chicken isolates, including one strain of serotype Schwarzengrund with a cluster of nine *pap* genes including regulatory and structural determinants, a Brancaster strain with the *papI* and *papB* regulatory genes without any structural gene, and a Johannesburg isolate with seven structural and accessory genes of the *pap* fimbriae but without the regulatory genes (Table 4).

### *Salmonella* pathogenicity islands (SPI) virulence factor distribution

Within *S. enterica* core genome, SPI-1 and 2 are ubiquitous as they contain key virulence factors necessary for host cell invasion and intercellular survival, respectively. All isolates collected from both human clinical and meat samples contained virulence factors within SPI-1, 2, and 3 to support these functions. SPI-3 genes *mgtCB* and *misL* which aid intracellular survival in low nutrient environments by increasing affinity for magnesium and perpetuate intestinal colonization, respectively, were present in 100% of isolates. All isolates also contained the SPI-4 genes *pipB* and *sopB* which are suspected to effect SPI-1 and 2 expression for epithelial

invasion, enteric salmonellosis, and chicken colonization. Pathogenicity virulence factors on other SPIs associated with invasive salmonellosis varied across several serotypes in this dataset (S2 Table). Typhoid toxin *cdtB* from SPI-11, proven to be significantly associated with high rates of invasive disease, was identified in 22% (16/72) isolates, 42% (8/19) clinical and 15% (8/53) chicken isolates (S2 Table). SPI-12 *sspH2*, conferring actin polymerization and enhanced cell-to-cell motility was found in 35% (25/72) isolates, 26% (5/19) clinical and 38% (20/53) chicken. SPI-24 genes *shdA* and *ratB* that propagate adhesion and host cell invasion also detected in 26% (19/72) and 21% (15/ 72) respectively (S2 Table).

## Discussion

Invasive NTS continues to be an emerging threat to public health, especially for regions throughout sSA where larger populations exist with depressed host immunity either from malnutrition, lack of sanitation, and/or immunosuppression form Malaria, HIV or other chronic illnesses. *S. enterica* disease severity is attributed to the combination of host immunity and the strain fitness (AMR and virulence factors often horizontally acquired within SPI and/or vertically through plasmids). This dataset shows a wide distribution of AMR and virulence factors supportive of systemic invasive salmonellosis within several different serovars, isolated from both human diarrheal patients and from chicken meat sold at a market in Dakar, Senegal.

### *Salmonella* transmission from chicken

In this study, we applied a genomics approach to analyze 19 human and 53 chicken isolates of *Salmonella enterica* collected during a surveillance project in Dakar, Senegal. We found a high serotype diversity in human isolates with 16 serovars represented in 19 strains. This observation shows the ability of many *Salmonella* serotypes to cause clinical infection in humans. Contrary to human strains, only 10 different serotypes were represented in the 53 chicken isolates analyzed with Brancaster (15 isolates), Kentucky (13 isolates), and Hadar (11 isolates) predominating in these strains. Noticeably, only serotypes Kentucky ST198 and Schwarzengrund were represented in both human (one isolate each) and chicken (13 and three isolates, respectively) strains. Chicken is an important source of *Salmonella* contamination to humans. Chicken products, including eggs, meat, and their derivative, are the cause of foodborne infection worldwide [43]. In Senegal, several studies reported *Salmonella* contamination of chickens, including retail meat from markets [15, 44], farm birds [45, 46] and cooked food [47, 48]. Many of the serotypes found in these studies can cause clinical infections in humans [15]. They are asymptomatically carried by chickens, which represent an important reservoir and a source of transmission. A few of these serotypes may cause mortality to newly hatched birds or egg loss but are well tolerated by adults [49]. They colonize different organs, including the reproductive tract, without compromising chicken growth and are thus vertically transmitted. Actually, the most threatening *Salmonella* serovar for poultry is Gallinarum, a chicken-restricted serotype that causes an acute systemic infection resulting in a high mortality rate [50]. To prevent losses, farmers heavily use antimicrobials to protect chicken against *Salmonella* [51]. In HIC, this concern is properly addressed by legislation and regulations applied to food manufacture and commercialization [52]. In contrast, such regulations do not exist in many LMIC, or when they exist, they are often not enforced. Additionally, poor hygiene is associated with food production in these countries, with street food being an important supply in many cities.

Although many studies have reported *Salmonella* contamination of chicken in Senegal, transmission from this source to humans has been rarely mentioned. This is primarily due to limited approaches in previous studies focused on species identification, periodic serotyping,

and antimicrobial susceptibility testing. One report in 2001 described a new serotype, Keur-massar, that was isolated from chicken meat and a human clinical sample in the same time period suggesting potential transmission through the food chain [15]. The WGS approach used in this study, by revealing the whole genome, is better suited to compare isolates from different sources and detect transmission of *Salmonella* or other pathogens from food to humans. Whole genome phylogenetic analysis clearly shows one human clinical isolate (serotype Schwarzengrund) that shares the same phylogenetic clade as three isolates procured from chicken meat (Fig 1). Similarly, out of the five *Salmonella* Kentucky of the ST198 that clustered together in phylogenetic analysis, one was isolated from the stools of a diarrheic patient while the other four originated from meat (Fig 1). Although this does not correspond to a demonstration of direct transmission from chicken to humans, it is a likely possibility. The case of Kentucky ST198 is important since this strain emerged in Egypt in 1989 after acquiring a genomic island that harbors several AMR genes [53]. Kentucky ST198 is disseminated widely in the world and causes clinical infections in humans. MDR phenotypes are concerning for public health.

## Animal reservoir of *Salmonella* and invasive nontyphoidal salmonellosis in sub-Saharan Africa

Besides chicken, *Salmonella* has been detected in several food animals in Senegal, including small ruminants [54], beef [55], and swine [56]. Additionally, *Salmonella* is well spread in Senegalese wildlife species and has been detected in vulture [57], bats [58], and turtles [59]. This endemic presence and the diversity of serovars that can cause clinical infection in humans represent an important public health concern in context of the rising frequency of iNTS in sSA [11]. Invasive NTS is a systemic disease with a high mortality rate caused by serotypes other than Typhi and Paratyphi that are restricted to the human host [10, 60]. This severe disease mostly affects children and immunocompromised adults, especially those infected with HIV [10]. Invasive NTS is also frequently associated with malaria [61]. Several reports have declared that *Salmonella* is becoming a leading cause of bacteremia in sSA [11]. Still, many laboratories in this region are not equipped to perform adequate surveillance and diagnostics; it is likely that many cases of iNTS are not reported. Therefore, there is an urgent need to establish a surveillance of *Salmonella*, especially invasive strains and trace their source of acquisition by humans.

## Spread of antimicrobial resistant determinants in food animals and the need for a surveillance system to prevent transmission to humans

AMR was rare in human isolates with 17/19 strains susceptible to the 22 antimicrobials tested. In contrast, 48/56 of the chicken isolates displayed resistance to at least one antimicrobial. Importantly, our WGS data were consistent with the AMR phenotype of the isolates except in five cases. Three of these cases consisted of chicken isolates harboring a *sul1* or *sul2* and a *dfrA* ARG while sensitive to sulfamethoxazole-trimethoprim combination (Table 1). We do not know the explanation of this observation. It might be due to a low level of expression of one or both of these ARGs. Another case corresponded to a Schwarzengrund strain that harbored a *tetA* efflux pump while sensitive to tetracycline (Table 1). The fifth case was a Kentucky strain that had a *sul1* gene only and was resistant to the sulfamethoxazole-trimethoprim combination without a *dfrA* gene. This strain might harbor a gene that confers resistance to trimethoprim, but that is unknown.

The high frequency of AMR isolates in chicken points to the use of antimicrobials in farming to prevent or treat infectious diseases and promote growth [45]. The extent of improper

use of antimicrobial is not known. However, family units of chicken farming for family consumption or small-scale commercialization are widespread in Senegal. This leads to a continuous selection of MDR strains that can enter the food chain. In this study, we found a significant level of resistance to antimicrobials used for the treatment of clinical salmonellosis, including sulfamethoxazole-trimethoprim combination and fluoroquinolones. Fluoroquinolones were introduced to treat *Salmonella* infections after resistance emerged against the historical first-line molecules chloramphenicol, ampicillin, and sulfamethoxazole-trimethoprim [20]. Earlier studies in Senegal reported a low level of resistance to fluoroquinolones [62, 63]. More recent studies show *Salmonella* fluoroquinolone resistance is increasing [14, 25, 26] which is consistent with our results. Subsequently, β-lactams have recently become the first-line treatment against severe *Salmonella* infections in Senegal. Available data indicate that β-lactam resistance is still rare in Senegal, which is also consistent with our findings. A robust and integrated surveillance system is needed to monitor the emergence of AMR pathogens in order to identify potential sources, implement countermeasures, and reduce the transmission of resistance determinants to humans in Senegal. Such a surveillance can use the WGS methodology presented in this study. It can be based on a one health approach in which comparative genomics of *Salmonella* from human, animal, food and environmental origins will permit to identify sources of contamination and of AMR dissemination. This will contribute to keep active molecules available for treatment of bacterial infections.

## Supporting information

**S1 Table. Replicons, antimicrobial resistance and virulence genes harbored by human and chicken isolates of *Salmonella*.**
(XLSX)

**S2 Table. Invasive salmonellosis-associated virulence genes harbored by non-ubiquitous *Salmonella* Pathogenicity Islands in human and chicken isolates.**
(XLSX)

## Acknowledgments

We would like to thank Adja Bousso Gueye for helping with the detection of antimicrobial resistance genes and Ousmane Sow for technical assistance. We are grateful to Idrissa Dieng for the visualization of the phylogenetic tree under R Studio.

## Author Contributions

**Conceptualization:** Yakhya Dieye, Paula J. Fedorka-Cray, Siddhartha Thakur.

**Formal analysis:** Yakhya Dieye, Dawn M. Hull, Cheikh Fall.

**Investigation:** Abdoul Aziz Wane, Lyndy Harden, Bissoume Sambe-Ba.

**Methodology:** Abdoul Aziz Wane, Lyndy Harden.

**Resources:** Abdoulaye Seck, Paula J. Fedorka-Cray, Siddhartha Thakur.

**Supervision:** Yakhya Dieye, Bissoume Sambe-Ba, Abdoulaye Seck, Siddhartha Thakur.

**Writing – original draft:** Yakhya Dieye.

**Writing – review & editing:** Yakhya Dieye, Dawn M. Hull, Cheikh Fall, Paula J. Fedorka-Cray, Siddhartha Thakur.

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
