## [Decision Letter · Decision Letter 0]

24 Jan 2022

PONE-D-21-24873Genomic comparison of clinical human to chicken meat Salmonella isolates in Senegal: broilers as a source of antimicrobial resistance and potentially invasive nontyphoidal salmonellosis infectionsPLOS ONE

Dear Dr. Dieye,

Thank you for submitting your manuscript to PLOS ONE. After careful consideration, we feel that it has merit but does not fully meet PLOS ONE’s publication criteria as it currently stands. Therefore, we invite you to submit a revised version of the manuscript that addresses the points raised during the review process.

The reviewers indicated some minor problems with the manuscript. Please reply to all and adapt the manuscript appropriately, highlighting the changes.

We look forward to receiving your revised manuscript.

Kind regards,

Patrick Butaye, DVM, PhD

Academic Editor

PLOS ONE

Journal Requirements:

2. Please upload a copy of Figure 2, to which you refer in your text on page 19. If the figure is no longer to be included as part of the submission please remove all reference to it within the text.

Reviewers' comments:

Reviewer's Responses to Questions

**Comments to the Author**

1. Is the manuscript technically sound, and do the data support the conclusions?

Reviewer #1: Yes

Reviewer #2: Yes

2. Has the statistical analysis been performed appropriately and rigorously? 

Reviewer #1: I Don't Know

Reviewer #2: N/A

3. Have the authors made all data underlying the findings in their manuscript fully available?

Reviewer #1: Yes

Reviewer #2: Yes

4. Is the manuscript presented in an intelligible fashion and written in standard English?

Reviewer #1: Yes

Reviewer #2: Yes

5. Review Comments to the Author

Reviewer #1: Title of Manuscript (PONE-D-21-24873): Genomic comparison of clinical human to chicken meat Salmonella isolates in Senegal: broilers as a source of antimicrobial resistance and potentially invasive nontyphoidal salmonellosis infections

1st Author: Yakhya Dieye, PhD

Review Comments

1. Throughout the manuscript you have used ‘antibiotics’ and ‘antimicrobials’ interchangeably. Best to use one consistently – I think ‘antimicrobials’ would be more suitable for this type of manuscript that compares clinical (human) and animal isolates

2. Page 74-75: Please revise sentence to read Food animals (poultry, cattle and pigs) and their products (including eggs, milk and pork) constitute ….

3. Line 371-372: You could add to that statement on these lines a further statement to read, ‘or where such regulations exist, they are often not enforced’

Overall verdict: The paper is well written and offers analysis of AMR from a somewhat one-health approach (has human and animal samples). Future similar analyses will be strengthened by incorporating environmental and plant samples in an expanded dataset. I recommend it for publication once the above minor corrections have been addressed.

Reviewer #2: In the present study, the authors performed whole genome sequencing of 72 Salmonella isolates recovered from humans (n=19) and chicken meat (n=53) during a surveillance project in Dakar, Senegal. The aim of their study was to determine the resistance profile of nontyphoidal Salmonella strains from clinical cases in humans, and the potential origin of these strains from retail chicken meat. Their results suggested possible transmission of the emerging multidrug resistant (MDR) Kentucky ST198 and serotype Schwarzengrund from chicken to human.

The title and the introductory part (L22-31) in the abstract are too wordy

Abstract

Line 37: cephalosporins are not historical first-line drugs

Lines 38-39: ‘..pointing to the concern posed by the excessive use of antimicrobials in farming.’: do the authors have data to sustain this statement, at least for Senegal?

Introduction

Lines 53-62: Introducing Salmonella and Samonella infections is too wordy

Line 65: The term “developing countries” is no longer preferred. Recommend using the World Bank classification of low- and middle-income countries. Same for “developed countries” (high-income countries). + Line 370 and 372

Lines 69-72: some other data are available (see hereafter)

Carey ME et al. The Surveillance for Enteric Fever in Asia Project (SEAP), Severe Typhoid Fever Surveillance in Africa (SETA), Surveillance of Enteric Fever in India (SEFI), and Strategic Typhoid Alliance Across Africa and Asia (STRATAA) Population-based Enteric Fever Studies: A Review of Methodological Similarities and Differences. Clin Infect Dis. 2020 Jul 29;71(Suppl 2):S102-S110. doi: 10.1093/cid/ciaa367. PMID: 32725221; PMCID: PMC7388711.

Nikiema MEM et al. Contamination of street food with multidrug-resistant Salmonella, in Ouagadougou, Burkina Faso. PLoS One. 2021 Jun 17;16(6):e0253312. doi: 10.1371/journal.pone.0253312. PMID: 34138936; PMCID: PMC8211238.

Pulford CV, et al. Stepwise evolution of Salmonella Typhimurium ST313 causing bloodstream infection in Africa. Nat Microbiol. 2021 Mar;6(3):327-338. doi: 10.1038/s41564-020-00836-1. Epub 2020 Dec 21. PMID: 33349664; PMCID: PMC8018540.

Kalonji LM, et al. Invasive Salmonella Infections at Multiple Surveillance Sites in the Democratic Republic of the Congo, 2011-2014. Clin Infect Dis. 2015 Nov 1;61 Suppl 4:S346-53. doi: 10.1093/cid/civ713. PMID: 26449951.

Kariuki S, Onsare RS. Epidemiology and Genomics of Invasive Nontyphoidal Salmonella Infections in Kenya. Clin Infect Dis. 2015 Nov 1;61 Suppl 4(Suppl 4):S317-24. doi: 10.1093/cid/civ711. PMID: 26449947; PMCID: PMC4596933.

Langendorf C et al. Enteric bacterial pathogens in children with diarrhea in Niger: diversity and antimicrobial resistance. PLoS One. 2015 Mar 23;10(3):e0120275. doi: 10.1371/journal.pone.0120275. PMID: 25799400; PMCID: PMC4370739.

…. and many more

i.e. type https://pubmed.ncbi.nlm.nih.gov/?term=Jacobs+J+and+salmonella+and+africa&sort=date

and

https://pubmed.ncbi.nlm.nih.gov/?term=Weill+FX+and+salmonella+and+africa&sort=date

Quoted in Ref 18

Please rephrase this part (Lines 69-72) according to the supplied information

Line 75: eggs and milk are not food animals but remained important vehicles of Salmonella, especially eggs

Lines 90-94: Still accurate for some parts of the world but banned in other parts (In 2006, the EU banned the use of antibiotics as feed additives for growth promotion). This sentence has to be nuanced

Line 103: which COULD promote

M&M

Line 120: Human Salmonella isolates dated from 2012-2013. Same period for the Isolates from chicken. Make it clear at line 123. It is clearly enounced at lines 175-176.

Lines 132-139: For antibiotics abbreviations try to use some internationally recognized ones such as http://bsacsurv.org/science/antimicrobials/

OR from EUCAST

https://www.eucast.org/fileadmin/src/media/PDFs/EUCAST_files/Disk_test_documents/Disk_abbreviations/EUCAST_system_for_antimicrobial_abbreviations.pdf

Interestingly, those abbreviations were only used later in Table 3

Line 142: MasterPureTM Gram Positive DNA Purification Kit for Gram negative bacteria

Results

Lines 178-180: these are repetitions of what was mentioned in the M&M section

Line 193: were absent in chicken Not appropriate: were not found in Salmonella isolates from chicken.

Line 197: yes, clearly, serotypes Kentucky and Virchow are known to be closely associated with poultry

Lines 272-273 + 420-421: Additionally, one isolate of Kentucky ST198 was resistant to sulfamethoxazole-trimethoprim while possessing a sul1 only without a dfrA gene. We made the same observation in S. maltophilia isolates where the dfrA gene was absent and where the dihydrofolate reductase was mutated and was conferring resistance to trimethoprim (not yet published, only in biorxiv)

Lines 274-278: would have be better if MIC to fluoroquinolone antibiotic had been performed

Lines 296-298: I do not know about the performance of plasmidSPAdes algorithm for assembling large plasmids from whole genome sequencing data, but the most detected plasmids were only from the Col plasmid family (high copy number and small in size <10kb). Could this affect the results for larger plasmids from the IncH or IncI or F? If yes, this should be mentioned in the Limitations section. PlasmidFinder also allows to find replicons from WGS data.

Discussion

Line 348: a local market in Senegal. Add the city name where chicken meat was sold

Lines 388-392: add reference i.e.

Line 401: add doi: 10.1136/bmjgh-2021-005659 to Ref 6

Line 424-427: Not sure family units use antibiotics as growth promotors. Therefore, the link between improper use of antibiotics (line 425), small-scale farming (426) and continuous selection of MDR strains (427) is unclear to me.

Do data on antibiotic consumption in human and animals exist in Senegal? If yes, authors should refer to those data in the discussion section, especially in lines 432-439.

6. PLOS authors have the option to publish the peer review history of their article (what does this mean?). If published, this will include your full peer review and any attached files.

Reviewer #1: **Yes: **Prof. Samuel Kariuki

Reviewer #2: No

---

## [Author Response · Author response to Decision Letter 0]

2 Feb 2022

Responses to reviewers

Dear Editor,

We are submitting a revised version of manuscript PONE-D-24873 entitled “Genomic comparison of clinical human to chicken meat Salmonella isolates in Senegal: broilers as a source of antimicrobial resistance and potentially invasive nontyphoidal salmonellosis infections”. We have made all the revisions requested. We thank both reviewers for their comments and suggestions that significantly improved the quality of our manuscript. Below are the responses to the academic editor and reviewers.

Responses to the academic editor

Done

2. Please upload a copy of Figure 2, to which you refer in your text on page 19. If the figure is no longer to be included as part of the submission please remove all reference to it within the text.

Figure 2 was mistakenly mentioned on page 19 of the submitted manuscript. It has been changed to “S2 Table”, page 19, lines 327, 329 and 332 of the revised manuscript.

Captions (SI Table and S2 Table) have been added for Supporting Information files at the end of the Discussion section, page 24, lines 438 and 440 of the revised manuscript.

The reference list has been updated. All the added citations are mentioned in the responses to the reviewers.

5. We have made the additional changes:

• Line 343 of the original manuscript, line 336 of the revised version: “form” changed to “from”.

Responses to Reviewer 1

1. Throughout the manuscript, you have used ‘antibiotics’ and ‘antimicrobials’ interchangeably. Best to use one consistently – I think ‘antimicrobials’ would be more suitable for this type of manuscript that compares clinical (human) and animal isolates

The changes requested by the reviewer have been made lines 77, 96 and 233 of the revised manuscript.

2. Lines 74-75: Please revise sentence to read Food animals (poultry, cattle and pigs) and their products (including eggs, milk and pork) constitute ….

The changes requested by the reviewer have been made lines 67-68 of the revised manuscript.

3. Line 371-372: You could add to that statement on these lines a further statement to read, ‘or where such regulations exist, they are often not enforced’

The changes requested by the reviewer have been made line 365 of the revised manuscript.

Overall verdict: The paper is well written and offers analysis of AMR from a somewhat one-health approach (has human and animal samples). Future similar analyses will be strengthened by incorporating environmental and plant samples in an expanded dataset. I recommend it for publication once the above minor corrections have been addressed.

We thank the reviewer for his valuable suggestions that contributed to improving the quality of our manuscript.

Responses to Reviewer 2

The title and the introductory part (L22-31) in the abstract are too wordy

The title and the introductory part of the abstract have been shortened according to the reviewer’s suggestion (lines 21-27 of the revised manuscript).

Abstract

Line 37: cephalosporins are not historical first-line drugs

The corresponding sentence has been modified line 33 of the revised manuscript, to correct the mistake identified by the reviewer.

Lines 38-39: ‘..pointing to the concern posed by the excessive use of antimicrobials in farming.’: do the authors have data to sustain this statement, at least for Senegal?

There are several publications reporting surveys on antimicrobial use African countries. In the original manuscript, we mention a reference (Ref 41; Ref 45 in the revised version) lines 423-425 in the Discussion section (lines 416-417 of the revised manuscript) that reported a survey in chicken farms. We have further evidences from a survey (unpublished) made in 2020 on behalf of the AMR Working Group of the Permanent Secretariate of the High National Counsel for Global health Security Agenda (to which we are a member). The survey mentioned the following among others:

• Importers of antimicrobials (AM) distribute molecules without compliance to existing recommendations

• There is no control of the distribution

• 37.5% of the veterinary pharmacists provide AM without prescription

• 60% of the veterinary doctors prescribe AM treatment without antimicrobials susceptibility testing

• 50% of the veterinary doctors prescribe human AM for animal treatment

• Only 53% of the veterinary doctors follow guidelines for AM use in animals

• Etc.

Introduction

Lines 53-62: Introducing Salmonella and Samonella infections is too wordy

The introduction on Salmonella has been shorten according to the reviewer’s suggestion (lines 49-54 of the revised manuscript).

Line 65: The term “developing countries” is no longer preferred. Recommend using the World Bank classification of low- and middle-income countries. Same for “developed countries” (high-income countries). + Line 370 and 372

The changes requested by the reviewer have been made lines 24, 25, 57, 363 and 365 of the revised manuscript.

Lines 69-72: some other data are available (see hereafter)

Please rephrase this part (Lines 69-72) according to the supplied information

We thank the reviewer for bringing to our attention these important data. We have reformulated the pointed section, lines 61-65 of the revised manuscript and added a few of the suggested references (refs 6-9 of the revised manuscript). 

Line 75: eggs and milk are not food animals but remained important vehicles of Salmonella, especially eggs

The changes requested by the reviewer have been made line 68 of the revised manuscript.

Lines 90-94: Still accurate for some parts of the world but banned in other parts (In 2006, the EU banned the use of antibiotics as feed additives for growth promotion). This sentence has to be nuanced

The sentence has been change lines 83-87 to incorporate the reviewer’s suggestion.

Line 103: which COULD promote

The changes suggested by the reviewer have been made line 96 of the revised manuscript.

M&M

Line 120: Human Salmonella isolates dated from 2012-2013. Same period for the isolates from chicken. Make it clear at line 123. It is clearly enounced at lines 175-176.

The changes suggested by the reviewer have been made line 114 of the revised manuscript.

Lines 132-139: For antibiotics abbreviations try to use some internationally recognized ones such as http://bsacsurv.org/science/antimicrobials/ OR from EUCAST https://www.eucast.org/fileadmin/src/media/PDFs/EUCAST_files/Disk_test_documents/Disk_abbreviations/EUCAST_system_for_antimicrobial_abbreviations.pdf. Interestingly, those abbreviations were only used later in Table 3

We do agree with the reviewer regarding the abbreviations used for antimicrobials. We used a two-letter code instead of the more commonly used three-letter code for a formatting purpose, in order to have all the information fit in Table 3. This Table is quite long and spans across three pages. The two-letter code helped minimize it length while avoid case breaking between pages.

Line 142: MasterPureTM Gram Positive DNA Purification Kit for Gram negative bacteria

This kit was used because it yielded from Salmonella genomic DNA of better quality for sequencing using an Illumina platform. This was added lines 137-139 in the revised manuscript.

Results

Lines 178-180: these are repetitions of what was mentioned in the M&M section

We deleted the sentences mentioned from the Results section.

Line 193: were absent in chicken Not appropriate: were not found in Salmonella isolates from chicken.

The changes suggested by the reviewer have been made line 186 of the revised manuscript.

Line 197: yes, clearly, serotypes Kentucky and Virchow are known to be closely associated with poultry

Agree.

Lines 272-273 + 420-421: Additionally, one isolate of Kentucky ST198 was resistant to sulfamethoxazole-trimethoprim while possessing a sul1 only without a dfrA gene. We made the same observation in S. maltophilia isolates where the dfrA gene was absent and where the dihydrofolate reductase was mutated and was conferring resistance to trimethoprim (not yet published, only in biorxiv)

We are very grateful to the reviewer for sharing this discovery. We would be very interesting to know more about the mentioned biorxix manuscript. We are trying to identify the genetic determinant supporting the resistance to trimethoprim and would be interested to collaborate with the reviewer in this topic. 

Lines 274-278: would have be better if MIC to fluoroquinolone antibiotic had been performed

We do agree with the reviewer. Unfortunately, a MIC system is currently not available in our laboratory to perform the suggested analysis.

Lines 296-298: I do not know about the performance of plasmidSPAdes algorithm for assembling large plasmids from whole genome sequencing data, but the most detected plasmids were only from the Col plasmid family (high copy number and small in size <10kb). Could this affect the results for larger plasmids from the IncH or IncI or F? If yes, this should be mentioned in the Limitations section. PlasmidFinder also allows to find replicons from WGS data.

We tested both plasmidSpades and PlasmidFinder and found the former to give results that are more reliable since it permits to get most of the plasmid sequences. In other genomics studies, we recovered sequences of large plasmids (the biggest being +300 kb) using plasmidSpades. We did not find any bias regarding plasmid size in the strains analyzed in this study and in others not published yet. We will be happy to share more on our experience using plasmidSpades.

Discussion

Line 348: a local market in Senegal. Add the city name where chicken meat was sold

The changes suggested by the reviewer have been made line 341 of the revised manuscript.

Lines 388-392: add reference i.e.

A reference (ref 53 of the revised version) has been added line 383 as requested by the reviewer.

Line 401: add doi: 10.1136/bmjgh-2021-005659 to Ref 6

The indicated article has been added as ref. 60 and mentioned line 394 of the revised manuscript.

Line 424-427: Not sure family units use antibiotics as growth promotors. Therefore, the link between improper use of antibiotics (line 425), small-scale farming (426) and continuous selection of MDR strains (427) is unclear to me.

In Senegal, broilers available to consumers come mostly from family units that raise chicken for consumption and for limited commercialization, and from small and mid-size chicken farms. There is no legislation banning the use of antimicrobials as growth promoter in farming, but only recommendations to which farmers are often not aware of. Consequently, empirical use of antimicrobials to promote animal growth is a common practice.

Do data on antibiotic consumption in human and animals exist in Senegal? If yes, authors should refer to those data in the discussion section, especially in lines 432-439.

Data on human and animal consumption of antimicrobial are not systematically collected and made available in Senegal. However, Permanent Secretariate of the High National Counsel for Global health Security Agenda, through its AMR Working Group is currently putting in place recommendations what would lead to regulations and legislations in the near future.

We would like to thank the reviewer for his thorough review and for the interesting suggestions and guidance he provided.

---

## [Editor Report · Decision Letter 1]

14 Mar 2022

Genomics of human and chicken Salmonella isolates in Senegal: broilers as a source of antimicrobial resistance and potentially invasive nontyphoidal salmonellosis infections

PONE-D-21-24873R1

Dear Dr. Dieye,

We’re pleased to inform you that your manuscript has been judged scientifically suitable for publication and will be formally accepted for publication once it meets all outstanding technical requirements.

Kind regards,

Patrick Butaye, DVM, PhD

Academic Editor

PLOS ONE
---

## [Editor Report · Acceptance letter]

16 Mar 2022

PONE-D-21-24873R1 

Genomics of human and chicken *Salmonella* isolates in Senegal: broilers as a source of antimicrobial resistance and potentially invasive nontyphoidal salmonellosis infections 

Dear Dr. Dieye:

I'm pleased to inform you that your manuscript has been deemed suitable for publication in PLOS ONE. Congratulations! Your manuscript is now with our production department. 

Kind regards, 

on behalf of

Professor Patrick Butaye 

Academic Editor

PLOS ONE